# CD74 Promotes a Pro-Inflammatory Tumor Microenvironment by Inducing S100A8 and S100A9 Secretion in Pancreatic Cancer

**DOI:** 10.3390/ijms241612993

**Published:** 2023-08-20

**Authors:** Woosol Chris Hong, Da Eun Lee, Hyeon Woong Kang, Myeong Jin Kim, Minsoo Kim, Ju Hyun Kim, Sungsoon Fang, Hyo Jung Kim, Joon Seong Park

**Affiliations:** 1Department of Medicine, Yonsei University College of Medicine, Seoul 03722, Republic of Korea; chrish95@gmail.com (W.C.H.); juhyun9503@gmail.com (J.H.K.); sfang@yuhs.ac (S.F.); 2Department of Surgery, Gangnam Severance Hospital, Yonsei University College of Medicine, Seoul 06273, Republic of Korea; hsgtj06321@gmail.com (D.E.L.); kanghw9305@gmail.com (H.W.K.); audwls8739@gmail.com (M.J.K.); alstndi777@gmail.com (M.K.); 3Brain Korea 21 PLUS Project for Medical Science, Yonsei University College of Medicine, Seoul 03722, Republic of Korea

**Keywords:** PDAC, CD74, inflammation, fibroblasts, tumor microenvironment

## Abstract

Pancreatic ductal adenocarcinoma (PDAC) is an aggressive form of pancreatic cancer with a poor prognosis and low survival rates. The prognostic and predictive biomarkers of PDAC are still largely unknown. The receptor CD74 was recently identified as a regulator of oncogenic properties in various cancers. However, the precise molecular mechanism of CD74 action in PDAC remains little understood. We investigated the role of CD74 by silencing CD74 in the pancreatic cancer cell line Capan-1. CD74 knockdown led to reductions in cell proliferation, migration, and invasion and increased apoptosis. Moreover, silencing CD74 resulted in the decreased expression and secretion of S100A8 and S100A9. An indirect co-culture of fibroblasts and tumor cells revealed that fibroblasts exposed to conditioned media from CD74 knockdown cells exhibited a reduced expression of inflammatory cytokines, suggesting a role of CD74 in influencing cytokine secretion in the tumor microenvironment. Overall, our study provides valuable insights into the critical role of CD74 in regulating the oncogenic properties of pancreatic cancer cells and its influence on the expression and secretion of S100A8 and S100A9. Taken together, these findings indicate CD74 as a potential diagnostic biomarker and therapeutic target for pancreatic cancer.

## 1. Introduction 

Pancreatic ductal adenocarcinoma (PDAC) is an aggressive cancer characterized by a poor prognosis and high mortality rates. Despite improvements in treatments, the incidence of pancreatic cancer is increasing, and it is ranked the seventh leading cause of cancer deaths worldwide [1,2]. Approximately 80–85% of pancreatic cancers are determined as unresectable at the time of diagnosis, contributing to the low patient survival rates [3]. In addition, PDAC is associated with a high metastatic capacity to adjacent organs such as the liver and lungs [4]. A key factor that contributes to tumor formation in PDAC is the tumor microenvironment (TME). The TME of pancreatic cancer consists of cancer-associated fibroblasts (CAFs), stellate cells, and various types of immune cells. Recent studies have shown that these cells play active roles in controlling tumor growth, invasion, and metastasis [5]. Furthermore, numerous pro-inflammatory cytokines, including IL-6, TNF-α, chemokine (C-X-C motif) ligand 1 (CXCL-1), and CXCL-6 are found in the TME of pancreatic cancer cells [6]. Various cells of the TME secrete such pro-inflammatory cytokines, creating conditions for chronic inflammation [7]. Therefore, the need to identify novel biomarkers that affect the TME is urgent for both diagnostic and therapeutic applications.

The cluster of differentiation 74 (CD74), also known as the invariant chain, is an MHC class II chaperone with a crucial role in antigen presentation. Recent studies have shown that CD74 in B-cell lymphomas and colorectal, lung, and breast cancers [8,9,10] serves as a receptor for the cytokine macrophage migration inhibitory factor (MIF) and the subsequent activation of signaling pathways [11]. The binding of MIF to CD74 is known to trigger several signaling pathways, such as the phosphoinositide-3-kinase (PI3K)/protein kinase B (Akt) pathway, and the activation of the transcription factor NF-ĸB [12]. The intracellular domain of CD74 (CD74-ICD) has also been implicated in the direct activation of NF-ĸB in B cells, suggesting that the receptor plays a role in inducing inflammation [13]. In addition, previous studies have demonstrated that the upregulation of CD74 expression is associated with increased inflammation and the secretion of pro-inflammatory cytokines in the TME [14,15]. However, the precise molecular mechanisms and functional significance of CD74 in pancreatic cancer have yet to be fully understood.

S100A8 and S100A9 are Ca^2+^-binding EF-hand (helix E–loop–helix F) proteins that belong to the S100 protein family. S100A8/A9 are secreted as heterodimers and bind to receptors such as TLR-4, RAGE, and CD36 [16,17]. The heterodimer acts as a chemoattractant to recruit inflammatory cells and create an inflammatory microenvironment that promotes tumor development [18]. In addition, S100A8/A9 have been shown to modulate tumor growth and metastasis in colon and breast cancers [19]. Moreover, S100A8/A9 are overexpressed in PDAC tissues and have been linked to a poor prognosis in patients [20]. These findings suggest that S100A8 and S100A9 play crucial roles in tumor progression within the TME.

In this study, we discovered that the activation of CD74 promotes oncogenic properties in pancreatic cancer and aids in the formation of an inflammatory TME. The knockdown of CD74 led to decreases in proliferation, invasion, and migration and increased apoptosis in vitro. We also demonstrate that CD74 regulates the expression and secretion of S100A8 and S100A9 via the TRAF6-NF-ĸB pathway, which has not been previously reported. We show that S100A8/A9 interact with fibroblasts to regulate the expression of pro-inflammatory cytokines, such as IL-6, CXCL1/6, and TNF-α. Lastly, we demonstrate that CD74 knockdown reduces tumor growth in vivo. Taken together, our results suggest that CD74 not only regulates the oncogenic properties in pancreatic cancer, but also has broader impacts as a TME regulator.

## 2. Results

### 2.1. CD74 Expression Is Elevated in Patients with PDAC

First, we aimed to confirm the elevated expression of CD74 in pancreatic cancer. An analysis using the gene expression profiling interactive analysis (GEPIA) database showed that the level of *CD74* mRNA in 179 cases of pancreatic cancer was significantly higher than that in 171 normal tissue samples (Figure 1A). The overall survival (OS) and disease-specific survival (DSS) of patients with high CD74 expression were lower than those of patients with low CD74 expression (Figure 1B). The TCGA data confirm that CD74 may serve as a prognostic marker of pancreatic cancer. Immunostaining for CD74 in normal tissues and tumor tissues showed an increase in positive CD74 staining in tumor tissues compared to that in normal tissues (Figure 1C). The data reveal strong disparities in CD74 expression between the normal samples and tumor samples, indicating that CD74 is highly expressed in pancreatic tumor tissues. The mRNA levels were examined in a panel of seven pancreatic tumors and compared to normal tissues using quantitative real-time PCR (qRT-PCR). The *CD74* expression in tumor tissues was greatly elevated compared to that in normal tissues (Figure 1D). The Western blot of five patients’ tumor tissues and normal tissues also showed a clear increase in CD74 expression in the former (Figure 1E).

We then tested the level of CD74 expression across five prominent PDAC cell lines: Capan-1, MIA-PaCa-2, BxPC-3, PANC-1, and AsPC-1. Interestingly, the evaluation of both the mRNA and protein CD74 levels showed that the expression was the highest in Capan-1 cells, with protein levels only being observed in Capan-1 cells (Figure 1F). Longer exposure of the Western blots revealed weak bands of CD74 in the other cell lines (Appendix A). We determined that CD74 should be highly expressed in the tested cell line and thus selected Capan-1 for our subsequent experiments. 

Our preliminary findings indicate that CD74 is highly expressed in pancreatic cancer tissues, that increased expression is associated with a worse prognosis for patients, and that the pancreatic cancer cell line Capan-1 expresses elevated levels of CD74.

### 2.2. CD74 Suppression Reduces Oncogenic Properties of Tumor Cells In Vitro

To explore the effects of CD74 suppression in pancreatic cancer cells, we performed an siRNA-mediated knockdown of CD74 in Capan-1 cells (Appendix A) and conducted RNA sequencing (RNA-seq) on the control and CD74 knockdown cells (Appendix A). The Gene Set Enrichment Analysis (GSEA) and gene ontology (GO) analysis showed that CD74-related genes were involved in cell adhesion, wound healing, the negative regulation of apoptosis, the positive regulation of cell migration, and the regulation of cytokines (Appendix A). 

To assess the effect of CD74 activation on cell proliferation, we performed a WST-1 viability assay on the CD74 knockdown cells (Figure 2A). We found that CD74 knockdown led to a markedly reduced growth rate in knockdown cells compared to that in the control cells. We then assessed the cell cycle distribution of the CD74 knockdown cells using flow cytometry. While the numbers of knockdown cells in the S and G2 phases were not significantly lower, substantially increased numbers of cells were observed in the G1 and sub-G1 phases, which indicate apoptotic and necrotic cells (Figure 2B). To confirm that CD74 plays an important role in the deregulation of apoptosis, we stained control and knockdown cells with Annexin V-FITC and propidium iodide (PI) and viewed the cells under a fluorescence microscope (Figure 2C). The merged results show an overall increase in staining for both Annexin V-FITC and PI in the knockdown compared to the control group, indicating that CD74 knockdown promoted apoptosis. Previous studies suggested that CD74 regulates the PI3K/Akt pathways [21]. In addition, CD74 signaling modulates the activity of hypoxia-inducible factor alpha (HIF-1α), leading to increased proliferation and survival [22]. Accordingly, we observed decreased expressions of p-AKT, p-mTOR, and HIF-1α, indicating that CD74 regulates cell proliferation by activating these pathways (Figure 2D). The effect of CD74 knockdown on cell proliferation and the regulation of apoptosis was also confirmed using PCR and Western blotting, as the expression of proliferative genes, such as *CDK4* and *CDK6*, decreased, and that of pro-apoptotic genes, such as *Bax*, *Caspase-9*, and *Caspase-3*, increased (Figure 2E). 

Next, we confirmed the role of CD74 in the migration and invasion of pancreatic cancer cells. We found that decreased CD74 expression led to a decreased migratory and invasive capacity of Capan-1 cells (Figure 2F,G). In accordance with the assay results, PCR and Western blotting revealed that the decrease in CD74 expression correlated with the increased levels of E-cadherin and N-cadherin, along with the decreased level of the migratory markers Snail and MMP-9 [23,24] (Figure 2H). Taken together, our results indicate that CD74 plays a crucial role in regulating oncogenic cell proliferation, migration, and invasion in pancreatic cancer.

### 2.3. CD74 Knockdown Decreases the Expression and Secretion of S100A8 and S100A9

We then explored the presence of potential downstream targets of CD74. Our gene ontology (GO) analysis indicated a decrease in the expression of genes that regulate cytokine secretion (Appendix A), suggesting that CD74 also regulates cytokine secretion in PDAC.

We first assessed the effect of CD74 knockdown on the expression of genes associated with cell migration, inflammatory responses, and secretion. We found that the expressions of cytokines *S100A8* and *S100A9* were considerably downregulated in the knockdown cells (Appendix A). *S100A8* expression was downregulated the most among the four common genes with a decreased expression in all three categories (Figure 3A). As mentioned before, the cytokines S100A8/A9 are associated with the induction of a pro-inflammatory microenvironment [25,26]. However, studies linking CD74-mediated regulation with S100A8/A9 expression or secretion are lacking. The Kyoto Encyclopedia of Genes and Genomes (KEGG) pathway analysis revealed that CD74 knockdown led to a reduced activation of the IL-17 signaling pathway and a decreased expression of S100A8/A9. A pathway analysis also showed that TRAF6 and NF-ĸB levels were reduced in the knockdown cells, leading to a decreased activation of the TRAF6-NF-ĸB signaling pathway (Appendix A). Although CD74 has been implicated in the activation of the NF-ĸB pathway [27], we did not find any previous studies on CD74-mediated activation of the TRAF6-NF-ĸB pathway. In accordance with our RNA-seq analysis, both the PCR and Western blot results clearly show that the levels of TRAF*,* TAK1*,* NF-ĸB, and S100A8/A9 were all reduced in the CD74 knockdown cells (Figure 3B,C). Furthermore, the enzyme-linked immunosorbent assay (ELISA) of the S100A8/A9 heterodimer secretion confirmed the reduced secretion in vitro (Figure 3D). Our results thus indicate that CD74 regulates the expression and secretion of S100A8 and S100A9.

We then investigated whether S100A8 and S100A9 could function as prognostic markers of pancreatic cancer. The OS of patients with a high S100A8 expression was significantly lower than that of patients with a low expression. We also found that the OS and DSS for patients with high CD74-high S100A8 expression were significantly worse than those in patients with low CD74-low S100A8, low CD74–high S100A8, and high CD74-low S100A8 expressions (Figure 3E). Therefore, our data indicate that both CD74 and S100A8 play crucial roles in determining the prognosis of patients with pancreatic cancer. We also observed that patients with a high S100A9 expression had a lower OS than patients with low expression; however, when we divided the patients into four categories according to S100A9 and CD74 expressions, the differences did not reach statistical significance (Appendix A). Next, we investigated whether S100A8/A9 were overexpressed in pancreatic tumor tissues. The GEPIA2 data indicate that both S100A8 and S100A9 were more highly expressed in the tumor issues than in the normal tissues (Figure 3F). Therefore, we assessed the relative mRNA and protein levels of S100A8/A9 in the patients’ tumor tissues and found that S100A8/A9 levels were significantly higher in the tumor tissues (Figure 3G,H). 

In summary, we demonstrate that CD74 regulates the expression and secretion of S100A8 and S100A9 and that these cytokines are clear prognostic markers in pancreatic cancer.

### 2.4. S100A8 and S100A9 Induce the Expression of Pro-Inflammatory Cytokines in Normal Fibroblasts (NFs)

Next, we examined the effects of the CD74-mediated secretion of S100A8 and S100A9 in the TME. S100A8/A9 have been implicated in the activation of inflammatory signaling pathways, and increased levels of S100A8/A9 were associated with chronic inflammation [28,29]. NFs and CAFs constitute significant proportions of the TME and are known to play substantial roles in tumor growth, extracellular matrix remodeling, and in promoting angiogenesis [30]. Recent reports have highlighted that the fibroblasts in the TME regulate and activate pro-inflammatory signaling pathways to promote inflammation [31]. Because the TME of PDAC is known to be densely populated by fibroblasts [32], we sought to examine whether the CD74-mediated secretion of S100A8/A9 induced the expression of inflammatory genes in isolated fibroblasts.

We first examined the protein level of α-SMA, which is known to be upregulated in CAFs, to confirm that the isolated fibroblasts were indeed NFs (Figure 4A). We then assessed changes in the expression of genes associated with inflammatory responses between NFs and CAFs using the RNA-seq data. The analysis indicated that the levels of pro-inflammatory cytokines, such as IL-6, CXCL-6, and CXCL-1, were upregulated in CAFs compared to NFs (Figure 4B). To test the effect of CD74 knockdown on the fibroblasts, we used conditioned media (CM) collected from control and siCD74 Capan-1 cells for the indirect co-culture experiment. After 72 h of CD74 knockdown in Capan-1 cells, the respective CM were collected, centrifuged, and used to treat NFs in six-well plates (Figure 4C). To determine whether the secretion of inflammatory cytokines by NFs was caused by S100A8/A9, we performed single and double knockdowns of S100A8 and S100A9 and collected CM (Figure 4D). We assessed the relative level of the S100A8/A9 heterodimer in each of the CM and found that the siS100A8, siS100A9, and siS100A8/A9 CM had lower heterodimer levels than the control media (Figure 4E). After confirming that heterodimer secretion decreased, we proceeded to use the CM collected from the knockdown cells.

After 72 h of indirect co-culture, we isolated the fibroblast RNA and analyzed the relative expressions of inflammatory cytokines via qRT-PCR. We found that the expressions of TNF-α, IL-6, CXCL-1, and CXCL-6 were significantly higher in the NFs grown in the control CM than in the NFs grown in the CD74-knockdown CM (Figure 4F). Several studies have indicated that the upregulation of HIF-1α expression induces the activation of inflammatory phenotypes in CAFs [33,34]. We observed a decrease in the expression of HIF-1α in fibroblasts cultured in siCD74 CM (Figure 4F), which suggests that HIF-1α may be activated by the S100A8/A9 signaling pathway. Furthermore, the expressions of TNF-α, IL-6, CXCL-1/6, and HIF-1α were reduced in the fibroblasts grown in the S100A8 and S100A9 knockdown CM (Figure 4G), indicating that S100A8/A9 play a direct role in inducing the expression of pro-inflammatory genes in fibroblasts. 

In conclusion, our co-culture results indicate that the CD74-mediated secretion of S100A8 and S100A9 induces inflammatory cytokine expression in fibroblasts, generating a highly inflammatory TME.

### 2.5. CD74 Knockdown Reduces Both Tumor Growth and Inflammatory Conditions in an Orthotopic Mouse Model

To observe the effects of CD74 on tumor growth in vivo, we constructed an orthotopic mouse model in BALB/C nude mice using Capan-1 cells. Capan-1 cells were directly injected into the pancreas of randomly selected mice, and the tumors were grown for eight weeks. The mice were then divided into two groups, and lentiviral particles containing shControl and shCD74 were injected. The mice were grown for one week after the viral injection and then euthanized (Figure 5A). A macroscopic observation revealed a smaller tumor size in the control mice than in the shCD74 mice. The tumor weight, too, significantly decreased in the shCD74 mice (Figure 5B) without a significant change in body weight. 

Furthermore, the tumor sections were harvested, and RNA was isolated for gene expression analysis. We found that the expressions of *CD74*, *S100A8/A9*, and *HIF-1*α were all reduced in the shCD74 group (Figure 5C). The reduced expression was further confirmed via immunofluorescence (IF) staining of CD74, S100A8/A9, HIF-1α, IL-6, and TNF-α to determine whether CD74 activation led to the secretion of inflammatory cytokines in the TME. Analyses of doubly stained CD74-S100A8 and CD74-S100A9 tumors revealed a decrease in the expression of both CD74 and S100A8/9 in shCD74 tumors compared to that of the control tumors (Figure 5D). We also observed lower levels of staining for HIF-1α, TNF-α, IL-6, and p-NF-ĸB in the shCD74 tumors compared to that of the control tumors (Figure 5E).

The in vivo results thus indicate that CD74 activation in pancreatic cancer tissues promotes tumor growth and incites a pro-inflammatory TME.

## 3. Discussion

PDAC is one of the most aggressive and lethal solid tumors, accounting for more than 90% of pancreatic cancer cases [35]. PDAC is characterized by high motility and inflammatory conditions, both of which are heavily influenced by the TME [36,37]. The TME of pancreatic cancer is diverse and plays significant roles in tumor progression, immune evasion, inflammation, and metastasis [32]. However, the lack of known TME biomarkers has hindered the development of novel treatments and therapeutics. We examined the molecular mechanism of CD74 action in pancreatic cancer cells and explored the broader impacts of CD74 activation on the TME. We investigated the role of CD74 activation in pancreatic cancer cells and its influence on oncogenic properties. In addition, we presented compelling evidence that CD74 regulates the secretion of cytokines S100A8 and S100A9, which, in turn, influence the formation of a pro-inflammatory TME. 

Recent studies reported on the high expression of CD74 in several types of tumor tissues and revealed that CD74 activation leads to increased proliferation and epithelial-to-mesenchymal transition and decreased apoptosis [8,12]. However, little is known regarding the role of CD74 in PDAC. Our patient data indicate that CD74 expression is upregulated in PDAC tissues and that a higher expression of CD74 leads to a worse OS. Moreover, suppressing CD74 expression in Capan-1 cells via siRNA-mediated knockdown led to decreases in wound healing and invasion. Using fluorescence-activated cell sorting (FACS) analysis and Annexin V-FITC and PI staining, we also revealed that CD74 knockdown decreases cell proliferation and increases apoptosis. CD74 is known to promote survival and cell proliferation through the PI3K/Akt signaling pathways [38]. Accordingly, we observed decreases in AKT and mTOR signaling in the knockdown Capan-1 cells. Our study verified the role of CD74 in PDAC as well as the molecular mechanisms induced by CD74 activation.

CD74 signaling has been linked to the promotion of inflammation [38,39]. A previous study in gliomas reported that a high expression of CD74 was positively associated with secretions of inflammatory cytokines [40]. This led us to hypothesize that CD74 activation may be positively associated with a pro-inflammatory TME. Previous studies have indicated that CD74 mediates the activation of NF-ĸB, a widely recognized marker of inflammation [13,41]. Using RNA-seq analysis, we discovered that the expression of the cytokines S100A8 and S100A9 decreased in the knockdown cells. CD74 knockdown led to decreases in both the expression and secretion of S100A8/A9 from Capan-1 cells. We also discovered that CD74 regulates the secretion of S100A8/A9 by activating the TRAF6–NF-ĸB pathway, which has not been previously described. The ELISA secretion assay confirmed that suppressing CD74 led to a decreased secretion of the S100A8/A9 heterodimer. As both S100A8/A9 are prognostic markers for patients with pancreatic cancer [20,42], with those with a higher expression having a poorer OS, we believe that the expression levels of both CD74 and S100A8/A9 can also be crucial prognostic markers in PDAC. 

Additionally, we linked CD74 activation to inflammation in the TME by conducting co-culture experiments with Capan-1 and NFs. Fibroblasts are integral members of the TME, and previous studies have indicated that CAFs incite inflammation [31,43]. While it has been known that PDAC cells activate fibroblasts in the TME [44], previous studies have not directly linked S100A8/A9 to fibroblast activation. Our results show a decreased expression of inflammatory cytokines from fibroblasts grown in the siCD74 CM compared to those grown in the control CM. Through the indirect co-culture of fibroblasts and tumor cells, we demonstrated that the CD74-mediated secretions of S100A8/A9 could induce oncogenic expressions of cytokines in NFs. In addition, we discovered an increase in the expression of HIF-1α in fibroblasts grown in the Capan-1 CM. As HIF-1α expression is known to be upregulated in CAFs compared to NFs [45], we believe that S100A8/A9 may play roles in fibroblast activation.

In summary, our study indicates that CD74 is a biomarker associated with tumor progression and inflammation in PDAC. High CD74 expression is correlated with increased proliferation, migration, and dysregulation of apoptosis. Moreover, CD74 signaling leads to the expression and secretion of cytokines S100A8 and S100A9 through the activation of the TRAF6–NF-ĸB pathway. In addition, we discovered that S100A8/A9 induce the secretion of pro-inflammatory cytokines in fibroblasts, promoting inflammation in the TME. These results may provide opportunities for future therapeutic interventions targeting CD74 and cell interactions in pancreatic cancer.

## 4. Materials and Methods

### 4.1. Patient Tissue Collection

This single-institution prospective study was approved by the Institutional Review Board (3–2015-0102), and written informed consent was obtained from each patient. Thirty patients diagnosed with pancreatic cancer at the Gangnam Severance Hospital from 2018 to 2019 underwent pancreatic resection.

### 4.2. Cell Lines and Cell Culture

PDAC cell lines BxPC-3, Capan-1, MIA PaCa-2, PANC-1, and AsPC-1 were purchased from the American Type Culture Collection (ATCC; Manassas, VA, USA). Cell lines were cultured in Roswell Park Memorial Institute medium 1640 (RPMI) (used for Capan-1, BxPC-3, and AsPC-1) or Dulbecco’s Modified Eagle’s Medium (DMEM) (used for MIA PaCa-2, and PANC-1), supplemented with 10% fetal bovine serum (FBS; Biowest, Riverside, MO, USA) and 1% antibiotic–antimycotic reagent (Gibco, Waltham, MA, USA) at 37 °C and 5% CO_2_. Specifically, Capan-1 cells were cultured in a 10 cm dish in the RPMI medium to 90% confluency before subculture every seven days. Medium was renewed every three days.

### 4.3. Western Blotting

Cells were harvested, washed with PBS, and lysed using RIPA buffer. Cell lysates were separated using SDS PAGE and transferred onto polyvinylidene fluoride membranes. After blocking with 5% skim milk, the membranes were incubated with the primary antibodies (1:1000) at 4 °C overnight, followed by incubation with the horseradish peroxidase (HRP)-conjugated secondary antibodies (1:5000) for 1 h. Development was performed using Clarity Western ECL Substrate (Bio-Rad, Gladesville, Australia) and detected on X-ray films (AGFA, Greenville, SC, USA) or using ImageQuant LAS 4000 (GE Healthcare, Chicago, IL, USA).

### 4.4. siRNA Transfection

For gene knockdown, small interfering RNAs (siRNAs) were purchased from Santa Cruz Biotechnology (Dallas, TX, USA). The siRNAs were dissolved in RNase-free H_2_O, and transfection was conducted using Lipofectamine RNAiMAX (Invitrogen, Paisley, UK) according to the manufacturer’s instructions. Cells were harvested and processed 48–72 h post-transfection.

### 4.5. Isolation and Culture of NFs and CAFs 

CAFs and NFs were isolated from human PDAC, and adjacent normal tissues were obtained from patients at the Gangnam Severance Hospital, Yonsei University (Seoul, Republic of Korea). Briefly, the fresh tissues were minced to ~1 mm^3^ and then digested with 1 mg/mL collagenase P (Sigma-Aldrich, St. Louis, MO, USA) at 37 °C for 1 h. After filtration with 70 μm cell strainers, the cells were collected via centrifugation at 1500 rpm for 5 min and plated with RPMI containing 10% FBS. After 3 d, the medium was replaced with fresh medium to remove non-adherent cells. All primary fibroblasts used in this study were at early passages (between 3 and 7).

### 4.6. Annexin V-FITC and PI Staining Assay

Capan-1 cells were seeded in 3.5 mm plates at a density of 3 × 10^5^ cells per plate and transfected with siRNA. After 72 h, the cells were washed twice with DPBS. Next, 5 µL of Annexin V-FITC and 5 µL PI were diluted in 500 µL 10× binding buffer, and the samples were stained for 15 min in the dark. The stained cells were analyzed under a fluorescence microscope. 

### 4.7. PI Staining Analysis

Capan-1 cells were seeded in 6-well plates at a density of 3 × 10^5^ cells per well and transfected with siRNA. The cells were harvested, fixed in 70% ethanol, and stained with PI (Sigma-Aldrich) and RNase A (Sigma-Aldrich) for 30 min in the dark. The fluorescence intensity was measured using an FACScanto II flow cytometer (BD Biosciences, Franklin Lakes, NJ, USA). A minimum of 10,000 events were collected for each sample. Cell cycle analysis of DNA histograms was performed using the FlowJo™ v10.8.1 software.

### 4.8. WST-1 Assay

Capan-1 cells were seeded in 96-well plates and transfected with siRNA. After incubation, the growth medium was replaced with 10% water-soluble tetrazolium-1 reagent (DoGenBio, Seoul, Republic of Korea). The absorbance of each well at 450 nm was measured using a VersaMax microplate reader (Molecular Devices, San Jose, CA, USA).

### 4.9. CM Preparation

Capan-1 cells were seeded in 6-well plates at a density of 3 × 10^5^ cells per well and transfected with siRNA. After 48 h, the medium was replaced with fresh serum-free medium. CM was collected and centrifuged for 10 min at 2000 rpm to remove cell debris.

### 4.10. ELISA

S100A8/S100A9 concentrations in the Capan-1 CM were measured using a Human Calprotectin ELISA kit (S100A8/S100A9) (ab267628; Abcam, Cambridge, MA, USA) according to the manufacturer’s instructions. Briefly, standards and samples were pipetted into 96-well-plate wells coated with primary antibodies against S100A8/A9. The wells were washed, and a biotinylated anti-calprotectin antibody was added. Next, HRP-conjugated streptavidin was added after a brief wash. After 45 min of incubation, TMB substrate solution was added to the wells and allowed to react. The optical densities of the wells at 450 nm were measured using the VersaMax microplate reader. The concentration of the S100A8/A9 heterodimer was calculated from the absorbance values using a standard curve.

### 4.11. RNA Isolation and PCR

Total RNA was isolated using the TRIzol reagent (Invitrogen). RNA quality was assessed on an Agilent TapeStation 4000 system (Agilent Technologies, Amstelveen, The Netherlands), and RNA quantification was performed using an ND-2000 Spectrophotometer (Thermo Fisher Scientific Inc., Wilmington, DE, USA).

After RNA isolation, first-strand cDNA synthesis was performed with 1 µg RNA as a template using the RT-qPCR cDNA Synthesis Kit (iNtRON Biotechnology, Seongnam, Republic of Korea) according to the manufacturer’s instructions. The amplification used the following conditions: initial denaturation at 95 °C for 5 min, followed by 30 cycles of denaturation at 95 °C for 30 s, annealing temperature for 30 s, and extension at 72 °C for 30 s. The amplification products were detected on a 2% agarose gel with ethidium bromide staining. The relative mRNA levels of targets were normalized to those of GAPDH, which served as the loading control.

### 4.12. qRT-PCR

qRT-PCR was performed using the StepOne™ Real-Time PCR System (Applied Biosystems, Foster City, CA, USA). The relative mRNA expression level was calculated using the 2^−ΔΔCT^ method, with GAPDH as the reference gene.

### 4.13. RNA-Seq Analysis

For control and test RNAs, library construction was performed using a QuantSeq 3′ mRNA-Seq Library Prep Kit (Lexogen, Inc., Vienna, Austria) according to the manufacturer’s instructions. In brief, each total RNA was prepared, and an oligo-dT primer containing an Illumina-compatible sequence at its 5′ end was hybridized to the RNA, followed by reverse transcription. After degradation of the RNA template, second-strand synthesis was initiated by a random primer containing an Illumina-compatible linker sequence at its 5′ end. The double-stranded library was purified using magnetic beads to remove all reaction components. The library was amplified to add the complete adapter sequences required for cluster generation. The final library was purified from PCR components. High-throughput single-end 75 sequencing was performed using NextSeq 550 (Illumina, Inc., San Diego, CA, USA).

### 4.14. Wound Healing Assay 

Capan-1 cells were seeded in 12-well plates and transfected with siRNA for 48 h. The cells were scratched using a 10 µL pipette tip and washed with DPBS, and microscope images of migrated cells were collected every 0, 16, and 24 h. The areas of each image were then measured using ImageJ.

### 4.15. Transwell Invasion Assay

Capan-1 cells were seeded in 6-well plates and transfected with siRNA for 48 h. The 8 µm pore size Transwell system (Corning Inc., Corning, NY, USA) was coated with Matrigel (1:50; Corning) for 1 h at room temperature. Next, 2 × 10^5^ transfected cells were seeded on the apical side of the Transwell chamber (24-well insert) in medium without FBS, and medium containing 10% FBS was added to the basal compartment as a chemoattractant. The cells were allowed to invade for 24 h. The cell numbers in each image were counted using ImageJ (Version 1.53t).

### 4.16. Mouse Tumor Model

BALB/c nude mice (6 weeks of age, male) were purchased from Oriente Bio. Capan-1 cells (3 × 10^6^) were mixed with serum-free DMEM and Matrigel (1:1) and injected into the pancreas. After eight weeks, the mice were divided into two groups of six and were injected with 2 × 10^6^ shControl or shCD74-expressing lentiviral particles (OriGene, Rockville, MD, USA). After two weeks, the mice were sacrificed, and the tumors were harvested, weighed, and fixed in 4% paraformaldehyde.

All animal experimental procedures followed the National Institutes of Health Guide for the Care and Use of Laboratory Animals and were performed in accordance with the protocols approved by the Institutional Animal Care and Use Committee of the Seoul Yonsei Pharmaceutical University Experimental Animal Center. 

### 4.17. IHC Staining

Serial sections (5 µm) of each block were adhered to poly-L-lysine-covered slides and incubated at 62 °C for 60 min. After deparaffinization and rehydration, the sections were heated in 10 mm citrate buffer (pH 6.0) for 15 min, and the slides were blocked with 10% goat serum in PBS for 1 h. The slides were washed with PBS and then incubated with the primary antibodies overnight. Alexa Fluor^TM^ 488 goat anti-mouse IgG (H + L) (Invitrogen; A11001) and Texas Red^®^-X goat anti-rabbit IgG (H + L) (Molecular Probes; T-6391) were used as secondary antibodies. The slides were viewed under an LSM 980 confocal microscope (Carl Zeiss AG, Oberkochen, Germany).

### 4.18. Statistical Analysis

Statistical analysis involved one-way or two-way ANOVA using GraphPad Prism version 8.0 (GraphPad Software, La Jolla, CA, USA). Data are presented as the mean ± standard deviation. Statistical significance is indicated (* *p* < 0.05; ** *p* < 0.01).

## Figures and Tables

**Figure 1 ijms-24-12993-f001:**
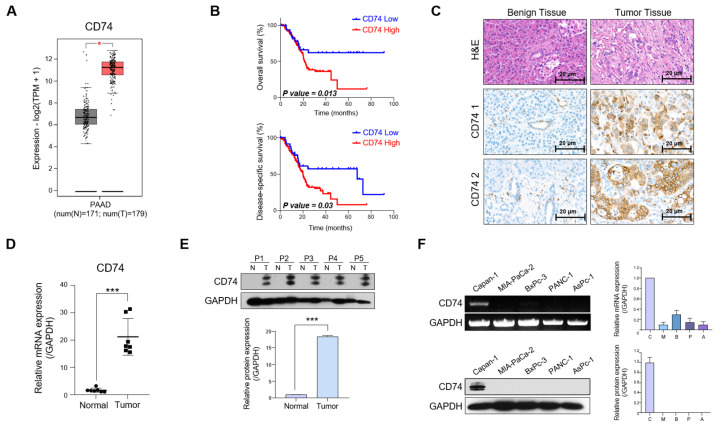
Elevated expression of CD74 in patients with PDAC. (**A**) CD74 expression in PDAC represented by GEPIA2 database. The gray represents expression level in the normal tissues and the red represents tumor tissues. (**B**) Kaplan-Meier curves of overall and disease-free survival of PDAC patients with high CD74 expression based on TCGA data. (**C**) Representative images of the IHC staining analyses for CD74 in benign and malignant tissues. (**D**) The mRNA expression of CD74 in normal and PDAC tissues via qRT-PCR. (**E**) The protein expression of CD74 in normal and PDAC tissues via Western blot. GAPDH expression was used as the control. (**F**) The mRNA and protein expression level of CD74 in five PDAC cell lines (Capan-1, MIA-PaCa-2, BxPc-3, PANC-1, and AsPc-1) via PCR and Western blot, respectively. * *p* < 0.05, *** *p* < 0.0001.

**Figure 2 ijms-24-12993-f002:**
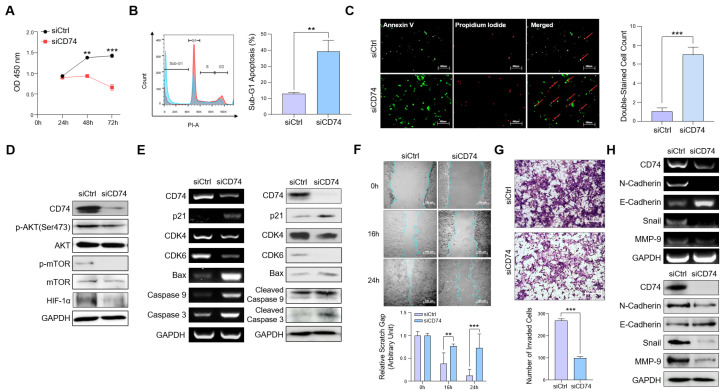
CD74 inhibition reduced PDAC cell proliferation and migration and promoted apoptosis. (**A**) WST-1 assay was used to analyze cell viability after CD74 siRNA treatment. (**B**) Propidium iodide (PI) staining was performed on control and siCD74 Capan-1 cells and analyzed via flow cytometry. The red represents the control cells while the blue represents the siCD74 cells. Scale bar = 500 μm. (**C**) PI and Annexin-V-FITC staining were detected via fluorescence microscopy. The red arrows represent the double-stained cells via PI and Annexin-V-FITC. (**D**) Protein expressions of p-AKT, AKT, p-mTOR, mTOR, and HIF-1α in control and siCD74 Capan-1 cells were estimated via Western blot. (**E**) The mRNA and protein expression of proliferation and apoptosis-related genes were estimated via PCR and Western blot. (**F**) The migration of control and siCD74 cells was measured with wound healing assay. Scale bar = 100 μm. (**G**) The number of invaded cells was measured with Transwell invasion assay. Scale bar = 100 μm. (**H**) The mRNA and protein expression of migration-related genes were estimated via PCR and Western blot. ** *p* < 0.01, *** *p* < 0.0001.

**Figure 3 ijms-24-12993-f003:**
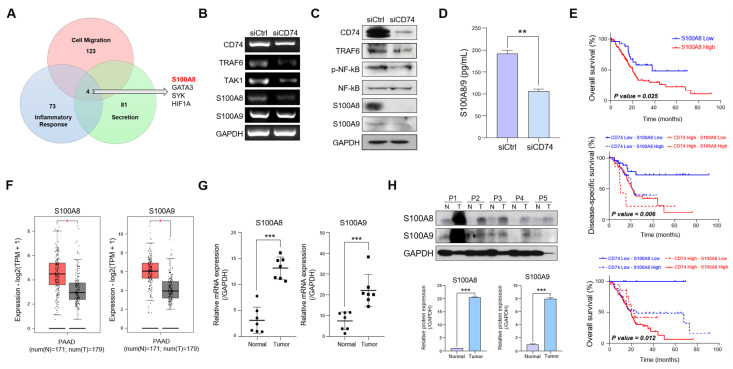
CD74 regulates expression and secretion of S100A8/A9 through the TRAF6-NFĸB pathway. (**A**) Venn diagram of differentially expressed genes in the cell migration, inflammatory response, and secretion-related genes in control and siCD74 Capan-1 cells. *S100A8* was the most downregulated gene among the categories. (**B**) The mRNA expressions of genes indicated in the TRAF6-NFĸB pathway in control and siCD74 Capan-1 cells were estimated via PCR. (**C**) The protein expression of indicated genes was estimated via Western blot. (**D**) Secretion of S100A8/A9 heterodimer in the conditioned medium of control and siCD74 Capan-1 cells was measured via ELISA analysis. (**E**) Kaplan–Meier curves of overall and disease-specific survival of PDAC patients with high S100A8, low CD74-low S100A8, low CD74–high S100A8, high CD74-low S100A8, and high CD74-high S100A8 based on TCGA data. (**F**) S100A8 and S100A9 expressions in GEPIA2 database. The red represents expression level in the tumor tissues while the gray represents the normal tissues. (**G**,**H**) The mRNA and protein expression of S100A8/A9 in normal and PDAC tissues via qRT-PCR and Western blot. * *p* < 0.05, ** *p* < 0.01, *** *p* < 0.0001.

**Figure 4 ijms-24-12993-f004:**
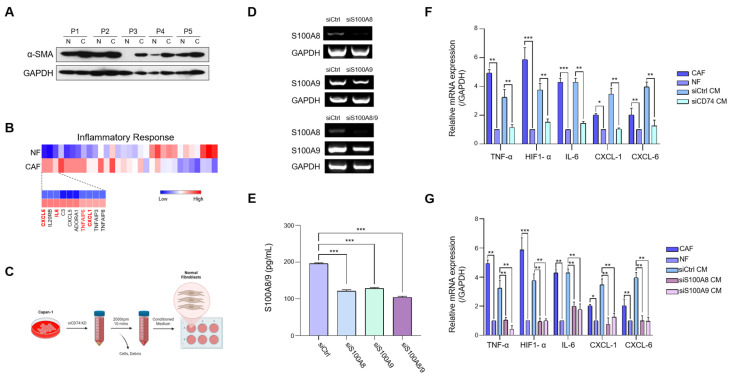
S100A8 and S100A9 induce expressions of pro-inflammatory cytokines in fibroblasts. (**A**) The protein expression of α-SMA in normal and cancer-associated fibroblasts (NFs and CAFs, respectively) was measured via Western blot. (**B**) Heatmap of differential expressions of inflammatory response genes of NFs and CAFs. (**C**) Schematic of the indirect co-culture assay. (**D**) Expressions of S100A8 and S100A9 in control, siS100A8, siS100A9, and siS100A8/A9 double-knockdown Capan-1 cells were measured via RT-PCR. (**E**) Secretion of S100A8/A9 heterodimer in the conditioned medium of control, siS100A8, siS100A9, and siS100A8/A9 double-knockdown Capan-1 cells was measured via ELISA analysis. (**F**) Expressions of TNF-α, HIF-1α, IL-6, CXCL-1/6 in CAFs, NFs, NF grown in control Capan-1-conditioned medium, and NF grown in siCD74 Capan-1-conditioned medium were measured via RT-qPCR. (**G**) Expressions of indicated genes in CAFs, NFs, NFs grown in control Capan-1-conditioned medium, and NFs grown in siS100A8 and siS100A9 Capan-1 conditioned medium were measured via RT-qPCR. * *p* < 0.05, ** *p* < 0.01, *** *p* < 0.0001.

**Figure 5 ijms-24-12993-f005:**
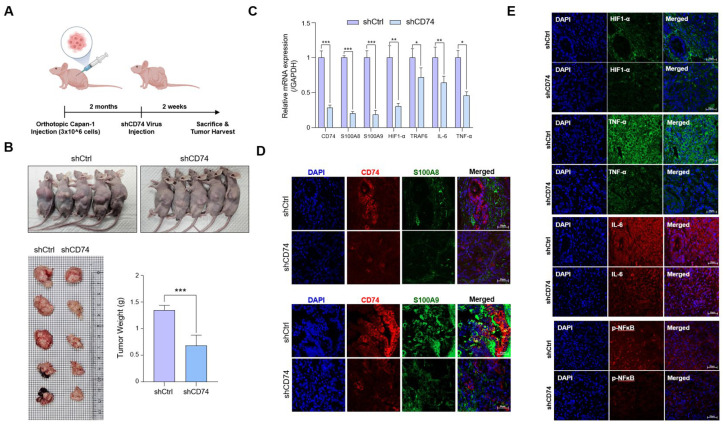
Reduced CD74 expression leads to reduced tumor growth and inflammatory conditions in orthotopic mouse models. (**A**) BALB/C nude mice were orthotopically injected with Capan-1 cells (3 × 10^6^ cells). After two months, lentivirus particles carrying shControl or shCD74 were injected (2 × 10^6^ particles). (**B**) Representative image of mice injected with control and shCD74 lentiviral particles, and image of tumors from the mice. The tumor weight was measured in grams. (**C**) Expressions of CD74, S100A8/A9, HIF-1α, and TRAF6, IL-6 in mice injected with control and shCD74 lentivirus were measured via RT-qPCR. (**D**) Representative images of immunofluorescence (IF) staining analyses of CD74-S100A8 and CD74-S100A9 double-stained tumors. (**E**) Representative images of IF staining analyses of HIF-1α, TNF-α, IL-6, and p-NFĸB. * *p* < 0.05, ** *p* < 0.01, *** *p* < 0.0001. Scale bar = 20 μm.

## Data Availability

The datasets used and analyzed in this paper are available from the corresponding authors upon reasonable request.

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
