# Peer review of "CD74 Promotes a Pro-Inflammatory Tumor Microenvironment by Inducing S100A8 and S100A9 Secretion in Pancreatic Cancer"

_ijms, 2023, doi:10.3390/ijms241612993_

Round 1
Reviewer 1 Report
In manuscript, CD74 Promotes a Pro-inflammatory Tumor Microenvironment by Inducing S100A8 and S100A9 Secretion in Pancreatic Cancer, Hong et al. presented a very interesting study. The authors clarified the regulatory axis of CD74 and S100A8/S100A9 in the context of PDAC tumor microenvironment.
In general, it’s a clear story and the manuscript was written in a relatively easy-following way. Authors provided good evidence to support the point they raised. However, some concerns should be properly solved or answered before this manuscript can reach a publish quality.
1. All the in vitro and in vivo data and was based on one PDAC cell line Capan-1, it’s hard to tell if CD74 regulating secretion of S100A8/S100A9 is cell line specific. Does it applicable to other cell lines? From figure S1A, we can still see CD74 are positively expressed other lines as well, the authors should at least verify it in another cell line like BxPC3, AsPC-1, or generate patient derived organoids. It’s surprising that CD74 are highly expressed in all the 5 patient tumors, but only one PDAC cell line.
2. Line 146: “migratory markers Snail and MMP-9”, need to provide reference to support Snail and MMP-9 are migratory markers.
3. Please provide higher resolution images for Fig 2C (Propidium Iodide column), Fig 2F and Fig S2B.
4. Please indicate the number of patients in Fig 3E, and the legend is missing in the bottom plot. There are more than two cohorts in some plots but only one p-value, what’s the p-value for? The use of OS(%) or DSS(%) in the Y axis should keep consistent in Fig 3E. And the authors didn’t explain how they got DSS ratio.
5. The figure legends of Fig 4F and G are very confusing. I assume the -/- mean NFs?
6. Line 465-467: “Capan-1 cells (3 × 106) were mixed with serum-free DMEM and Matrigel (1:1) and injected into 466 the pancreas.” It sounds impossible for me to inject such huge amount of cells into pancreas orthotopically, wondering how the authors made it? Can you explain how much volume you injected into the mouse pancreas? Fig 5A is not a schematic for orthotopic injection but subcutaneous injection. Please provide higher resolution images for Fig 5B as well as tumor growth curve. From Fig 5B we can tell ethically, it seems some of the tumors already exceeded the 2cm limit.
Reviewer 2 Report
A study "CD74 Promotes a Pro-inflammatory Tumor Microenvironment
by Inducing S100A8 and S100A9 Secretion in Pancreatic Cancer" by Woosol Chris Hong investigates a role of CD74 in regulating the oncogenic properties of pancreatic cancer cells and its influence on the expression and secretion of S100A8 and S100A9. The manuscript is written well, the results are of good quality and conclusive. Conclusions are based on the results presented. The results have been also appropriately discussed.
Specific comments:
1. The Authors should more broadly discussed why the majority of cell lines (Fig. 1F) do not express CD74, which is in certain contrast to other results in this figure. I also recommend to show the figure included in S1A to demonstrate mRNA level of CD74.
2. Fig. 2C - the bar graph and its content are unclear. What is shown in y-axis?
3. In general, the figures are barely visible. I would recommend to enlarge the majority of the panels, especially when free space is available.
Round 2
Reviewer 1 Report
Reviewer #1 Comments:
1. All the in vitro and in vivo data and was based on one PDAC cell line Capan-1, it’s hard to tell if CD74 regulating secretion of S100A8/S100A9 is cell line specific. Does it applicable to other cell lines? From figure S1A, we can still see CD74 are positively expressed other lines as well, the authors should at least verify it in another cell line like BxPC3, AsPC-1, or generate patient derived organoids. It’s surprising that CD74 are highly expressed in all the 5 patient tumors, but only one PDAC cell line.
A) Thank you for your comment. While Figure S1A does show that CD74 is expressed across all cell lines, the bands of MIA-PaCa-2, BxPc-3, PANC-1, and AsPc-1 were obtained after an exposure time of over one hour. Because our principal goal of the investigation was to study the function of CD74 by suppressing its expression, we believed it would be reasonable to use Capan-1, which displayed a strong CD74 expression, for our experiments.
The authors didn’t indicate where did the five patient samples in figure 1E come from. All the five tumors are highly expressed with CD74. If there’s not a 2nd PDAC cell line available to interrogate the function of CD74, why not isolate PDAC organoids from these patient samples? Are these 5 patient tumors selected? If not, can the authors explain why only one cell line out of a few cell lines expresses high level of CD74, but all the patient tumors do?
2. Line 146: “migratory markers Snail and MMP-9”, need to provide reference to support Snail and MMP-9 are migratory markers.
A) Thank you for your comment. We have provided references [23, 24] to support our statement in the Revised Manuscript.
Zheng, X.; Carstens, J. L.; Kim, J.; Scheible, M.; Kaye, J.; Sugimoto, H.; Wu, C. C.; LeBleu, V. S.; Kalluri, R., Epithelial-to-mesenchymal transition is dispensable for metastasis but induces chemoresistance in pancreatic cancer. Nature 2015, 527, (7579), 525-530.
Tang, D.; Zhang, J.; Yuan, Z.; Zhang, H.; Chong, Y.; Huang, Y.; Wang, J.; Xiong, Q.; Wang, S.; Wu, Q.; Tian, Y.; Lu, Y.; Ge, X.; Shen, W.; Wang, D., Correction: PSC-derived Galectin-1 inducing epithelial-mesenchymal transition of pancreatic ductal adenocarcinoma cells by activating the NF-κB pathway. Oncotarget 2021, 12, (20), 2111-2113.
3. Please provide higher resolution images for Fig 2C (Propidium Iodide column), Fig 2F and Fig S2B.
A) Thank you for your comment. We have updated higher resolution images for the figures in the Revised Manuscript.
I don’t see improvement of Fig 2C. No PI signal can be visualized in the PI channel, and in the merged images, there are comet like lines or arrows that are not explained.
4. 
Please indicate the number of patients in Fig 3E, and the legend is missing in the bottom plot. There are more than two cohorts in some plots but only one p-value, what’s the p-value for? The use of OS(%) or DSS(%) in the Y axis should keep consistent in Fig 3E. And the authors didn’t explain how they got DSS ratio.
A) Thank you for your comment. First, we have added the legend in the bottom plot in the Revised Manuscript.
Second, the p-value in the multiple cohort graphs reflects the Log-Rank test for the trend across the four cohorts.
Lastly, we obtained the DSS ratio based on TCGA data.
5. The figure legends of Fig 4F and G are very confusing. I assume the -/- mean NFs?
A) 
Thank you for your comment. Yes, the (-/-) group is the NF only group. We apologize for the confusion and have revised the legend in the Revised Manuscript. We would like to express our appreciation for pointing out our error.
6. Line 465-467: “Capan-1 cells (3 × 106) were mixed with serum-free DMEM and Matrigel (1:1) and injected into 466 the pancreas.” It sounds impossible for me to inject such huge amount of cells into pancreas orthotopically, wondering how the authors made it? Can you explain how much volume you injected into the mouse pancreas? Fig 5A is not a schematic for orthotopic injection but subcutaneous injection. Please provide higher resolution images for Fig 5B as well as tumor growth curve. From Fig 5B we can tell ethically, it seems some of the tumors already exceeded the 2cm limit.
A) Thank you for your comment. To prepare the cells for injection, we counted and resuspended the appropriate number of cells (3 × 106) in 45 µL PBS, then added 45 µL of Matrigel. We then injected 80 µL of the mixed cell and solution into the pancreas of the mice.
Second, we have updated the schematic and added higher resolution images for Figure 5B in the Revised Manuscript. However, we did not measure a tumor growth curve for our experiment as tumor growth measurements are not possible for orthotopic models as intraperitoneal tumors cannot be measured in the early stages. Therefore, we measured the final tumor volume.
Lastly, while the tumors do reach the 2cm limit, we believe that they do not exceed the limit. Thank you for your kind and considerate comments.

It will be extremely hard to inject 80ul of cells into one spot of the pancreas, and from figure 5B it looks like the cells were injected into the pancreas area, but also leaked to other regions. Can you provide the picture of pancreas with tumors when you open the mice for tissue collection?
Round 3
Reviewer 1 Report
No.